# ROBUST MULTI-AGENT REINFORCEMENT LEARNING DRIVEN BY CORRELATED EQUILIBRIUM

## ABSTRACT

In this paper we deal with robust cooperative multi-agent reinforcement learning (CMARL). While CMARL has many potential applications, only a trained policy that is robust enough can be confidently deployed in real world. Existing works on robust MARL mainly apply vanilla adversarial training in centralized training and decentralized execution paradigm. We, however, find that if a CMARL environment contains an adversarial agent, the performance of decentralized equilibrium might perform significantly poor for achieving such adversarial robustness. To tackle this issue, we suggest that *when execution* the non-adversarial agents must jointly make the decision to improve the robustness, therefore solving *correlated equilibrium* instead. We theoretically demonstrate the superiority of correlated equilibrium over the decentralized one in adversarial MARL settings. Therefore, to achieve robust CMARL, we introduce novel strategies to encourage agents to learn correlated equilibrium while maximally preserving the convenience of the decentralized execution. The global variables with mutual information are proposed to help agents learn robust policies with MARL algorithms. The experimental results show that our method can dramatically boost performance on the SMAC environments.

## 1 INTRODUCTION

Recently, reinforcement learning (RL) has achieved remarkable success in many practical sequential decision problems, such as Go (Silver et al., 2017), chess (Silver et al., 2018), real-time strategy games (Vinyals et al., 2019), etc. In real-world, many sequential decision problems involve more than one decision maker (i.e. multi-agent), such as auto-driving, traffic light control and network routing. Cooperative multi-agent reinforcement learning (CMARL) is a key framework to solve these practical problems. Existing MARL methods for cooperative environments include policy-based methods, e.g. MADDPG (Lowe et al., 2017), COMA (Foerster et al., 2017), and value-based methods, e.g. VDN (Sunehag et al., 2018), QMIX (Rashid et al., 2018), QTRAN (Son et al., 2019). However, before we actually apply CMARL's policy into real world applications, a question must be asked: are these learned policies safe or robust to be deployed? What will happen if some agents made mistakes or behaved adversarially against other agents? It will be most likely that the entire team might fail to achieve their goal or perform extremely poorly. (Lin et al., 2020) demonstrates the unrobustness in CMARL environment, where a learnt adversarial of one agent can hugely decrease the team's performance.

Therefore, in practice, we expect to have a multi-agent team policy in a fully cooperative environment that is robust when some agent(s) make some mistakes and even behave adversarially. To the best of knowledge, very few existing works on this issue mainly use vanilla adversarial training strategy. Klima et al. (2018) considered a two-agent cooperative case, in order to make the policy robust, agents become competitive with a certain probability during training. Li et al. (2019) provided a robust MADDPG approach called M3DDPG, where each agent optimizes its policy based on other agents' influenced sub-optimal actions.

Most state-of-the-art MARL algorithms utilize *centralized training and decentralized execution* (CTDE) routine, since this setting is common in real world cases. The robust MARL method M3DDPG also followed the CTDE setting. However, existing works on team mini-max normal form or extensive form games show that if the environment contains an adversarial agent, then the

decentralized equilibrium from CTDE routine can be significantly worse than the correlated equilibrium. We furthermore extend this finding into stochastic team mini-max games. Inspired by this important observation, if we can urge agents to *learn a correlated equilibrium* (i.e. the non-adversarial agents jointly make the decision when execution), then we may achieve better performance than those following CTDE in robust MARL setting.

In this work, we achieve the robust MARL via solving correlated equilibrium motivated by the latent variable model, where the introduction of a latent variable across all agents could help agents jointly make their decisions. Our contributions can be summarized as follows.

- We demonstrate that in stochastic team mini-max games, the decentralized equilibrium can be arbitrarily worse than correlated one, and the gap can be significantly larger than in normal or extensive form games.
- With this result, we point out that learning correlated equilibrium is indeed necessary in robust MARL.
- We propose a simple strategy to urge agents to learn correlated equilibrium, and show that this method can yield significant performance improvements over vanilla adversarial training.

## 2 RELATED WORKS

**Robust RL**  The robustness in RL involves the perturbations occurring in different cases, such as state or observation, environment, action or policy, opponent's policy, etc. **1)** For the robustness to state or observation perturbation, most works focused on adversarial attacks of image state/observation. Pattanaik et al. (2018) used gradient-based attack on image state, and vanilla adversarial training was adopted to obtain robust policy; Fischer et al. (2019) first trained a normal policy, and distilled it on adversarial states to achieve robustness; Ferdowsi et al. (2018) applied adversarial training to autonomous driving tasks that interfered agent's input sensors based on environment, and then conducted adversarial training; **2)** For the robustness to environment, robust Markov decision process (MDP) could be used to formulate this problem. Many works (e.g. Wiesemann et al. (2013); Lim et al. (2013)) have studied this model and provided both theoretical analysis and algorithmic design. In deep RL scenario, Rajeswaran et al. (2016) used Monte Carlo approach to train agent, while Abdullah et al. (2019); Hou et al. (2020) adopted adversarial training to obtain a robust agent to all environments within a Wasserstein ball. Mankowitz et al. (2019) conducted adversarial training in MPO algorithm to optimize the performance in the worst performance environment. **3)** To be against the perturbation of action or policy, Tessler et al. (2019); Gu et al. (2018); Vinitsky et al. (2020) considered the case that agent's action may be influenced by another action, and conducted adversarial training. **4)** For the robustness to opponent, Pinto et al. (2017); Ma et al. (2018) focused on the case that agent's reward may be influenced by another agent, and adversarial training was implemented to solve the two-agent game to obtain a robust agent.

**Correlated Equilibrium**  Correlated equilibrium is a more general equilibrium in game theory compared to Nash equilibrium. In a cooperative task, if the team agents jointly make decisions together, then the optimal team policy is correlated equilibrium. Correlated equilibrium is widely studied in game theory (e.g. Hart & Mas-Colell (2001; 2000); Neyman (1997)). In team mini-max game, solving the team's correlated equilibrium in normal form game is straightforward (just treat the team as a single agent); Celli & Gatti (2017); Zhang & An (2020); Farina et al. (2018) proposed various algorithms to solve correlated equilibrium in extensive formal games. In deep RL scenario, Celli et al. (2019) applied vanilla hidden variable model to solve correlated equilibrium in simple repetitive environments, while information loss with hidden variable model was used in Chen et al. (2019) to solve correlated equilibrium in normal multi-agent environment.

## 3 ROBUST MARL

### 3.1 BACKGROUND

A typical cooperative MARL problem can be formulated as a stochastic Markov game $(\mathcal{S}, \{\mathcal{A}_i\}_{i=1}^n, r, P)$, where $\mathcal{S}$ denotes the state space, $\{\mathcal{A}_i\}$ is $i$-th agent's action space. The en-

vironment starts at state $s_0$ based on some initial distribution $p_0$. At each time step $t$, agents select a joint action $\mathbf{a}_t \in \times_{i=1}^n \mathcal{A}_i$ based on some policy $\pi_{team}(\mathbf{a}_t|s_t)$, and receive reward $r(s_t, \mathbf{a}_t)$. The environment will transfer to a new state $s_{t+1} \sim P(\cdot|s_t, \mathbf{a}_t)$. The goal is to maximize agents' expected accumulated reward: $\max_{\pi_{team}} \mathbb{E}_{s_0 \sim p_0, \mathbf{a}_t \sim \pi_{team}, s_{t+1} \sim P} [\sum_t r(s_t, \mathbf{a}_t)]$.

Most state-of-the-art MARL algorithms utilize CTDE routine, that is, each agent selects its own action independently. If the environment is fully observable, then at each timestep $t$, each agent selects an action $a_{it} \in \mathcal{A}_i$ based on some policy $\pi_i(a_{it}|s_t)$, and if the environment is partially observable, then at each timestep $t$, each agent will receive an observation $o_{it}$ based on $s_t$, and select action based on policy $\pi_i(a_{it}|o_{it})$. And the goal becomes $\max_{\pi_{1:n}} \mathbb{E}_{s_0 \sim p_0, a_{1:n,t} \sim \pi_{1:n}, s_{t+1} \sim P} [\sum_t r(s_t, a_{1:n,t})]$.

### 3.2 ROBUST MARL AND VANILLA ADVERSARIAL TRAINING

The motivation of our work is to obtain a policy that is robust when one agent makes some mistakes. In normal MARL algorithm, the team can guarantee to achieve high reward only when all agents accurately execute their optimal strategies. However, this may not always be true in real world scenarios. Real world agents may occasionally make mistakes (e.g. machine malfunctioning). To achieve the robustness to this kind of mistakes, we propose to solve the worst case mini-max problem:

$$\text{Fixed } i \text{ or } \forall\, i, \ \max_{\pi_{1:n}} \min_{\pi_{i,mis}} \mathbb{E}_{s_0 \sim p_0, a_{it} \sim \pi_{i,mis}, a_{-i,t} \sim \pi_{-i}, s_{t+1} \sim P} \left[ \sum_t r(s_t, a_{1:n,t}) \right] \tag{1}$$
$$\text{s.t. } D(\pi_{i,mis}||\pi_i) \leq \varepsilon$$

where $\{-i\}$ means all except $i$. $\pi_{i,mis}$ means the mistaken policy that the mistaken agent actually perform. $D$ is some kind of distance measure, since we cannot expect that the team policy can still be robust when one agent makes very big mistakes.

Unfortunately, this mini-max problem is hard to solve, since it contains two MDPs nested with each other. On the other side, the common case in real world is that, agents make mistakes *randomly*, i.e. these mistakes are most likely not related to the team's goal and other agents' policies. Also, agents typically make mistakes *only occasionally*, since agents that always/frequently makes mistakes are not allowed to be deployed in practice. So following these considerations, we consider a simpler case and instantiate the robust cooperative MARL into QMIX algorithm.

We consider a weaker worst case mini-max problem. Since we assume agent only make mistakes occasionally, we consider the case that the mistaken agent executes its "worst" action in a certain probability $\varepsilon$. Also we assume that the mistakes are most likely not related to the team's goal and other agents' policies. In QMIX, if an agent $i$ doesn't consider other agents' policies, then its worst action is the one that minimizes its own $Q_i$ function (since in QMIX $\frac{\partial Q_{tot}}{\partial Q_i} \geq 0$, lower $Q_i$ will lead to lower $Q_{tot}$). In summary, we let the mistaken agent $i$ perform

$$a_{i,mis} = \begin{cases} \arg\max_a Q_i(s,a) & \text{with prob. } 1 - \varepsilon \\ \arg\min_a Q_i(s,a) & \text{with prob. } \varepsilon \end{cases} \tag{2}$$

and apply vanilla adversarial training to obtain a robust policy. The detailed algorithm framework 2 is described in Appendix A.1 . The performance of vanilla adversarial training will be considered as a baseline.

## 4 ROBUST MARL WITH CORRELATED EQUILIBRIUM

### 4.1 ROBUST MARL REQUIRES CORRELATED EQUILIBRIUM

In this part, we will show that with naive adversarial training in a centralized training and decentralized execution fashion, the learned policy might be sub-optimal in adversarial settings, thus requiring a correlated equilibrium.

In typical MARL settings, if the environment is fully cooperative, then the algorithms with centralized training and decentralized execution (e.g. QMIX/MADDPG/COMA, etc) can achieve state-of-the-art performance in certain environments. This indicates that at least for these environments,

correlation in execution is not necessary. Furthermore, Lauer & Riedmiller (2000) proved that decentralized executed policy can achieve optimal performance for fully observable and fully cooperative RL.

However, in the robust MARL setting, some agent(s) become adversarial, indicating that the environment is not fully cooperative now. So the question is: whether the correlation in execution is necessary in adversarial scenario?

For the settings of Eq. (1), the problems actually become a team mini-max game. The works on team mini-max normal form or extensive form games (von Stengel & Koller, 1997; Basilico et al., 2016; Celli & Gatti, 2017), pointed out that the decentralized equilibrium can be significantly worse than correlated equilibrium at least for some games. In normal form team mini-max game, Basilico et al. (2016) proved that the gap between correlated and decentralized equilibrium is at most $O(m^{n-2})$, where $n$ is the number of agents (includes the adversarial) and $m$ is each agent's number of action. And the bound is tight.

During this section, we simply define the "correlated equilibrium" as the equilibrium of optimal "correlated policy", that is, the team learns the optimal policy $\pi^*_{team}(\mathbf{a}|s_t)$ together, and "decentralized equilibrium" as the equilibrium of optimal "decentralized policy" learned by CTDE algorithm.

We denote $\mathbb{E}_{cor}$ as the team's expected reward under their optimal correlated policy, and $\mathbb{E}_{dec}$ as that in their optimal decentralized policy. In MARL, agents play a stochastic game. Since a repeated normal form game is a special case of stochastic games, so $\frac{\mathbb{E}_{cor}}{\mathbb{E}_{dec}}$ can be at least $m^{n-2}$ in stochastic games. Moreover, we find that this gap can be even larger in stochastic games than that in normal form games, because stochastic game is a sequential game, and agents' previous actions can influence the future state and therefore affect the future reward.

**Proposition 1.** *There exists a stochastic game that $\frac{\mathbb{E}_{cor}}{\mathbb{E}_{dec}} > m^{n-2}$.*

*Proof.* Consider this example: $\mathcal{S} = \{S_1, S_2\}$, initial state is $S_1$. Agents $1 \cdots n-1$ is a team, $n$ is adversary. $\mathcal{A}_i = \{1, \cdots, m\}$, $i = 1 \cdots n$. Discount factor $\gamma < 1$. The team's reward function $r(S_1, \mathbf{a}) = 0$, $\forall \mathbf{a}$. $r(S_2, \mathbf{a}) = \begin{cases} 1 & a_1 = \cdots = a_n \\ 0 & \text{otherwise} \end{cases}$. Deterministic state transition function $T(S_1, \mathbf{a}) = \begin{cases} S_2 & a_1 = \cdots = a_n \\ S_1 & \text{otherwise} \end{cases}$; $T(S_2, \mathbf{a}) = \begin{cases} \text{Game ends} & a_1 = a_2 = \cdots = a_n \\ S_2 & \text{otherwise} \end{cases}$.

In each state, the team's optimal correlated policy is to perform correlated action $\{1, \cdots, 1\}, \cdots, \{m, \cdots, m\}$ with equal probability $\frac{1}{m}$. Because if the team perform any of these actions with probability less than $\frac{1}{m}$, the adversary will perform that action, which will reduce the team's reward.

However, if the team plays in a decentralized way, then each agent's optimal policy is to perform each $m$ actions with equal probability $\frac{1}{m}$.

We can prove that in this example, $\frac{\mathbb{E}_{cor}}{\mathbb{E}_{dec}} \geq m^{2n-4}(1-r)^2$. The detailed derivation can be found in Appendix A.2. □

In fact, the gap between correlated and decentralized equilibrium in stochastic team mini-max game can be arbitrarily larger than normal form game, elaborated in the following proposition. The detailed derivation can be found in Appendix A.2.

**Proposition 2.** $\forall$ *fixed $k \in \mathbb{Z}_+$, there exists a stochastic game in which $\frac{\mathbb{E}_{cor}}{\mathbb{E}_{dec}} \geq O(m^{k(n-2)})$.*

Therefore, in robust MARL settings, CTDE algorithms may no longer achieve optimal performance. To achieve better robustness in adversarial settings, we need to design some methods that can urge agents to learn correlated equilibrium .

### 4.2 LEARNING CORRELATED EQUILIBRIUM

In this work, we propose a novel approach to learn correlated equilibrium for robust MARL agents: using global random variable with mutual information.

This method is inspired by the idea of InfoGAN (Chen et al., 2016). The idea is to add a global random variable $z$ as an extra input of the Q network, i.e. changing $Q_i(o_{it}, a_{it})$ to $Q_i(o_{it}, z_t, a_{it})$. Although the global random variable itself is meaningless, agents may learn to perform correlated equilibrium based on its value. The intuition is that, taking the example in Proposition 1, if we add a global random variable $z$ sampled from $[0, 1]$ uniformly at each timestep, then each agent might learn to perform action 1 when $z \in \left[0, \frac{1}{m}\right)$, perform action 2 when $z \in \left[\frac{1}{m}, \frac{2}{m}\right)$, and so on. Therefore, the overall policy is the optimal correlated policy of the team.

The following proposition is straight forward using the definition of correlated equilibrium. We also give a simple derivation in Appendix A.2.

**Proposition 3.** *For a fully observable, finite discrete action MARL environment, if all agents receive a global continuous random variable $z$, then there exists a deterministic policy $\mu_i(s, z) : \mathcal{S} \times \mathbb{R} \to \mathcal{A}_i, i \in \{1, \cdots, n\}$ equivalent to the team's optimal correlated (stochastic) policy $\pi^*(a_1, \cdots, a_n | s)$.*

We now summarize two advantages of the proposed global random variable method for learning correlation equilibrium in robust MARL settings.

- We can allow agents to perform correlated equilibrium while maximally preserving the property of CTDE. Since the global latent variable is independent of states, the only elements that agents need to share is a random number generator and a random seed. This can be done before the game starts, and agents can still perform "decentralized" policy during the game.

- In MDP or fully observable fully cooperative correlated multi-agent MDP, one can prove that there must exist an optimal deterministic policy (Puterman, 2014), and therefore deterministic policy algorithm can learn optimal policy. However, this property will not hold if the the multi-agent MDP is not fully cooperative, the optimal correlated policy in the example in Proposition 1 must be stochastic. This will prevent deterministic policy algorithms from learning optimal correlated policy directly. But when agents shares a global random variable, deterministic policy approaches will be possible to learn a policy that is equivalent to the optimal correlated policy.

To avoid the ignorance of the global random variable's information for each agent, we propose to maximize the mutual information between the random variable and the agent's action $I(z_t; a_{it})$, and therefore agents must use the information from $z$. Also, to avoid solving the posterior $P(z|a)$ directly, the variational lower bound is derived as an approximate objective, like InfoGAN and Barber & Agakov (2003) (The subscript $i$,$it$ is omitted):

$$I(z, a) \geq \mathbb{E}_{z,a}\left[\log \frac{q(z|a)}{p(z)}\right] = \mathbb{E}_{z,a}\left[\log q(z|a)\right] + H(z). \tag{3}$$

where $q(z|a)$ is the variational approximation of $P(z|a)$.

Unlike InfoGAN using latent variables $c$ to model the information that is irrelevant of generator source $z$ (width, italic), in reinforcement learning with latent variable, we prefer applying the latent variables to model the correlated policy that is related to current observations. Thus, instead of simply utilizing $I(z_t; a_{it})$, it's better to use the conditional mutual information $I(z_t; a_{it}|o_{it})$.

A similar variational lower bound can also be derived to approximate this conditional mutual information (The subscript $i$,$it$ is omitted):

$$I(z, a|o) = \mathbb{E}_{z,a,o}\left[\log \frac{p(z|a, o)}{p(z|o)}\right] = \mathbb{E}_{z,a,o}\left[\log \frac{q(z|a, o)}{p(z|o)}\right] + \mathbb{E}_{a,o}[D_{KL}(p(z|a, o)||q(z|a, o))]$$

$$\geq \mathbb{E}_{z,a,o}\left[\log \frac{q(z|a, o)}{p(z|o)}\right] = \mathbb{E}_{z,a,o}\left[\log q(z|a, o)\right] + H(z) \quad (\text{Since } p(z|o) = p(z)) \tag{4}$$

Since the entropy term $H(z)$ is a constant, thus ignored, we can define $\mathcal{L}_I = -\sum_{i=1}^{n} \mathbb{E}_{o_i, a_i \sim \mathcal{D}, z \sim p(z)}\left[\log q\left(z \mid a_i, o_i\right)\right]$, and use $\mathcal{L}_{tot} = \mathcal{L}_{RL} + \lambda_I \mathcal{L}_I$ as the overall loss function. In experiments, we follow InfoGAN's idea, configure variational approximation $q(z|\cdot)$ as a Gaussian

distribution, and apply a neural network to output its mean and variance. Algorithm 1 describes the overall training procedure.

---

**Algorithm 1:** Overall training procedure

---

Initialize replay buffer $\mathcal{D} = \emptyset$
**for** *each epoch* **do**
    **for** *sampling loop* **do**
        Obtain current $s_t$ or $\mathbf{o}_t$.
        Sample $z_t$ using some distribution.
        Rollout action $\mathbf{a}_t$ (using $\varepsilon$-greedy or other method, with $Q_i(s_t, z_t, a_{it})$ or
          $Q_i(o_{it}, z_t, a_{it})$).
        Select $i$ and rollout the mistaken action $a_{it,mis}$ using Eq. 2. Let
          $\mathbf{a}_{t,mis} = (a_{1t} \cdots a_{i-1,t}, a_{it,mis}, a_{i+1,t} \cdots a_n)$
        Perform action $\mathbf{a}_{t,mis}$ and get $r_t$, $s_{t+1}$ or $\mathbf{o}_{t+1}$.
        Store transition $(s_t, z_t, \mathbf{a}_{t,mis}, r_t, s_{t+1})$ or $(s_t, \mathbf{o}_t, z_t, \mathbf{a}_{t,mis}, r_t, s_{t+1}, \mathbf{o}_{t+1})$ to $\mathcal{D}$.
    **end**
    **for** *training loop* **do**
        Sample a minibatch $\mathcal{M}$ from replay buffer $\mathcal{D}$.
        Compute QMIX loss $\mathcal{L}_{TD}$.
        Compute overall loss $\mathcal{L}_{tot} = \mathcal{L}_{TD} + \lambda_I \mathcal{L}_I$ with $\mathcal{L}_I$ described above.
        Perform a update step of QMIX with loss $\mathcal{L}_{tot}$.
    **end**
**end**

---

**Comparison with existing works.** Both Chen et al. (2019) and Kim et al. (2020) involve the idea of global random variable and mutual information. However, both of these two works aim to improve coordination in standard MARL tasks, while we focus on solving robust CMARL. To the best of our knowledge, we are the first to demonstrate the importance of correlation in this kind of robust MARL setting. So as a first-step work, we just apply the most straightforward method to demonstrate this important finding:

- In our work, we formulate the global random variable as certain common knowledge to be shared between agents, which is straightforward, and do not need to specify its distribution as Chen et al. (2019) ($p_z(z)$ in Theorem 2.2 ).

- Kim et al. (2020) use the mutual information between each pair of agent's policy $I\left(\pi^i\left(\cdot \mid s_t\right); \pi^j\left(\cdot \mid s_t\right)\right)$ to encourage agent to coordinate. This is a much complicate method. As a first-step work, we suggest to employ conditional mutual information $I(z; a|s)$ (or $I(z; a|o)$) to encourage agents to use information from $z$, and allow agents to learn the correlated method themselves. Chen et al. (2019) proposed similar mutual information settings, but since they formulated $p_z(z)$, then the mutual information $I\left(z; \boldsymbol{\pi}^S(\boldsymbol{a}, z \mid s)\right)$ over joint distribution was used.

Whether the more complicated correlation method can achieve even better performance in robust CMARL settings is still an open question.

## 5 EXPERIMENTS

We test our method with QMIX algorithm in SMAC (Samvelyan et al., 2019) environment. We follow the experiment settings in QMIX (Rashid et al., 2018). The only difference is we train and evaluate with agents' mistaken action as in Eq. (2). We test two robust settings like Eq. (1), including: 1) only one fixed agent will make mistakes; 2) all agents can randomly make mistakes, but in each timestep at most one agent can make mistakes.

**Environment and Network Architecture** We use the SMAC environment involving decentralised micromanagement scenarios, where each unit of the game is controlled by an individual RL agent. It is a partially observable environment where each agent's observation is within a circu-

lar area around it. The action space is discrete. For further details of the environment, please refer to (Samvelyan et al., 2019).

In the training, we use a 3-dimensional independent uniform distribution $U(0, 1)$ as global variable, and set MI loss coefficient $\lambda_I = 0.1$. These are just hyperparameters, and we choose them based on performance.

We uses the same experiment settings of QMIX paper, including the architecture of the Q network, all the hyper-parameter settings and the performance evaluation method (evaluate every 10K timesteps with 32 games).

For the global variable part, to add latent variable $z$ into the model, we just extend the agent's observation space. For the mutual information part ($q(z|a, o)$), we uses an independent fully connected network with 2 32-unit-layers and ReLU activation function. The network takes agent's observation (without latent variable) and agent's action as input, and output a 6-dimensional vector with first 3 dimensional as mean and last 3 dimensional as variance. During training, we gather all agents' observation and action data together and feed them into this network together.

**Map and Agent Selection**    To evaluate our method, we select the maps and agents such that the original QMIX can achieve good performance in non-robust settings, but its policy is not robust with the selected agents.

By examining several maps about QMIX's performance and robustness, we choose four maps to evaluate our method: 8m, 2s3z, 3m, 3s5z. The detailed information can be found in Appendix A.3. In all the evaluations, we choose $\varepsilon = 0.3$ for 3m, and $\varepsilon = 0.5$ for other 3 maps.

We evaluate the "random agent" case in all except 3s5z maps. Because in 3s5z map, normal policy will still obtain 68% of winning rate when a random agent has 50% probability to select its worst action. For the "fixed agent" case, since many agents in these maps are homogeneous, we only select some representative agents for evaluation. The selection criteria is mainly based on the robustness of the normal QMIX policy, and the detailed information can be found in Appendix A.3. We evaluate a total of 9 different agent settings.

**Adversarial Testing Results**    In this part we compare 4 different settings: **n**ormal **p**olicy (train without adversarial but test with adversarial), **v**anilla **a**dversarial training (baseline), adversarial training with **g**lobal **v**ariable but without mutual information loss, adversarial training with **g**lobal variable and **m**utual information loss. In the whole section, we use the following abbreviations to indicate these four settings: NP, VA, GV, GM.

We run each experiment 5 times like QMIX does, and report the 25% and 75% as the upper and lower error bar. The mean winning rate during training is shown in Figure 1. We also plot the mean winning rate of the last few steps (in dotted line), and the performance of NP is plotted as black dotted line. The plots show that by adding global variable and mutual information loss, the performance of GM in most settings is better than VA and GV in average. The performance of GV is sometimes but not always better than VA. The reason might be that, without mutual information loss, agents will become easier to ignore global variable's information. Additional experimental results can be found in Appendix A.4.

In adversarial settings, the testing variance is larger than normal settings. So in order to evaluate the performance more accurately, we select the high checkpoint of each run within the error bar, and test it with 1000 episodes. Also, we test the model with different percentages of adversarial rate. The results are depicted in Figure 2, and further show that training with global variable and mutual information can lead to a better adversarial testing performance. The raw data can be found in Appendix A.5 in Table 3.

**Additional Baselines of Partially Observable Environments**    In Section 4 we use some examples to demonstrate the importance of correlation in robust CMARL settings for fully observable environments, where correlation is not important in normal CMARL settings but plays an key role in *robust* CMARL settings.

However, since SMAC is a partially observable environment, and some existing works (e.g. Bernstein et al. (2009)) pointed out that providing a global random variable to all agents can actually

improve the performance in partially observable environments. So in order to demonstrate that correlation is more important in robust CMARL settings, we build another baseline in the following.

We train a policy in normal settings with a global variable and mutual information loss (we use abbreviation NG to indicate it), and then evaluate it in adversarial settings. We compare the average performance improvement of NG/NP and GM/VA. The results can be found in Table 1 here and Table 2 in Appendix A.4.

The results shows that, in most maps and agents, the performance improvement of NG/NP is less than GM/VA. This indicates that "correlation is more important in robust settings" to a certain extent: in the same evaluation settings, the improvement of normal settings (normal training) is less than robust settings (adversarial training).

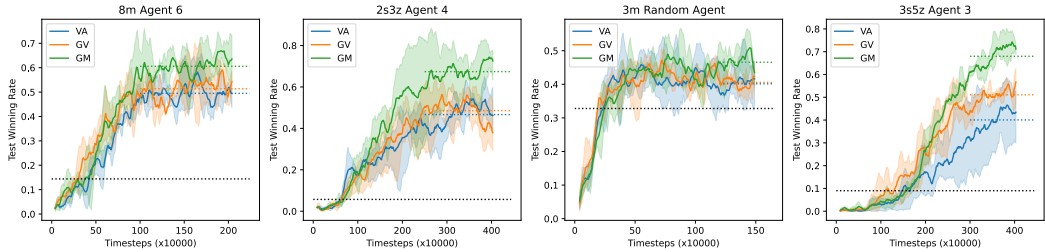

Figure 1: Test winning rate v.s. training steps for different maps and adversarial agents (NP: normal policy; VA: vanilla adversarial training; GV: adversarial training with global variable; GM: adversarial training with global variable and mutual information loss)

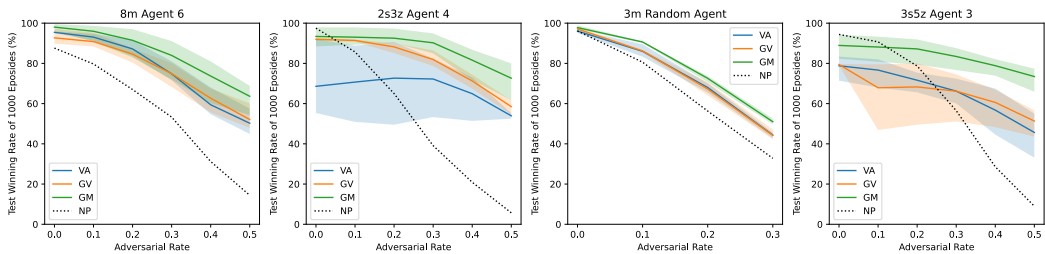

Figure 2: Adversarial testing results with different adversarial rates.

Table 1: Comparison of NG/NP and GM/VA: Mean test winning rate of different settings (NP: normal policy; NG: normal training with global variable and mutual information loss; VA: vanilla adversarial training; GM: adversarial training with global variable and mutual information loss)

| Map and Agent | 8m Agent 6 | 2s3z Agent 4 | 3m Random Agent | 3s5z Agent 3 |
|---|---|---|---|---|
| Adv Eval Rate | 0.5 | 0.5 | 0.3 | 0.5 |
| NP | 14.4 | 5.7 | 32.8 | 9.0 |
| NG | 25.1 | 8.0 | 40.7 | 15.8 |
| Gap | 10.7 | 2.3 | **7.9** | 6.8 |
| VA | 50.2 | 53.9 | 44.4 | 45.7 |
| GM | 63.6 | 72.6 | 51.0 | 73.5 |
| Gap | **13.4** | **18.7** | 6.6 | **27.8** |

**Random Testing Results** Apart from testing the adversarial agents, we also test scenarios that agent makes non-adversarial mistakes, i.e. agent plays random action with certain probability:
$$a_{mis} = \begin{cases} \arg\max_a Q(s, a) & \text{with prob. } 1 - \varepsilon \\ \text{Random choice from } \mathcal{A} & \text{with prob. } \varepsilon \end{cases}$$
. We use the same $\varepsilon$ as above.

Figure 3: Random testing results with different random rates.

The results are depicted in Figure 3, which show that training with global variable and mutual information can lead to a slightly better random testing performance. The raw data can be found in Appendix A.5 in Table 4.

## 6 CONCLUSION AND DISCUSSIONS

In this work, we focus on robust CMARL when one agent in the team makes mistakes or even behave adversarially. We found that in team mini-max stochastic games, the performance of decentralized equilibrium can be significantly worse than correlated equilibrium. Thus in order to achieve a better robust CMARL performance, we propose a method which uses global variables with mutual information to help agents learn correlated equilibrium. The experimental results show that this method can achieve better performance compared to vanilla adversarial training.

Robust CMARL is a important research direction, as it will directly determine whether we can safely deploy CMARL into practical problems. To the best of our knowledge, this research direction is still in the early stage. In this work, we just provide a simple method to solve the robust CMARL problem. Several future directions can be explored:

- The experiments of this work show that in some settings, the robust policy will slightly decrease the non-robust performance. So one possible future work is to balance the performance and robustness.

- We only consider a weak worst case mini-max problem. Whether it is possible to solve the real adversarial case (i.e. Eq. (1)) remains a future work.

- Besides, we only focus on the robustness of one agent. In reality, perhaps all agents can have a certain probability to make mistakes. Therefore, another possible future work is to consider this case.

- Also, as mentioned in the end of Section 4, we just use the most straightforward correlation method to demonstrate the importance of correlation in robust CMARL. Whether more complicated correlation methods can achieve even better performance in robust CMARL settings is still an open problem.

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

## A  APPENDIX

### A.1  VANILLA ADVERSARIAL TRAINING FRAMEWORK

---
**Algorithm 2:** Vanilla adversarial training framework

---
Initialize replay buffer $\mathcal{D} = \emptyset$
**for** *each epoch* **do**
  **for** *sampling loop* **do**
    Obtain current $s_t$ or $\mathbf{o}_t$, and rollout action $\mathbf{a}_t$ (using $\varepsilon$-greedy or other method).
    Select $i$ and rollout the mistaken action $a_{it,mis}$ using Eq. 2. Let
    $\mathbf{a}_{t,mis} = (a_{1t} \cdots a_{i-1,t}, a_{it,mis}, a_{i+1,t} \cdots a_n)$
    Perform action $\mathbf{a}_{t,mis}$ and get $r_t$, $s_{t+1}$ or $\mathbf{o}_{t+1}$.
    Store transition $(s_t, \mathbf{a}_{t,mis}, r_t, s_{t+1})$ or $(s_t, \mathbf{o}_t, \mathbf{a}_{t,mis}, r_t, s_{t+1}, \mathbf{o}_{t+1})$ to $\mathcal{D}$.
  **end**
  **for** *training loop* **do**
    Sample a minibatch $\mathcal{M}$ from replay buffer $\mathcal{D}$.
    Compute QMIX loss $\mathcal{L}_{TD}$.
    Perform a update step of QMIX with loss $\mathcal{L}_{TD}$.
  **end**
**end**

---

### A.2  DERIVATION OF PROPOSITIONS

**Detailed derivation of Proposition 1**  In correlated equilibrium, every step in $S_1$ has $\frac{1}{m}$ probability to transfer to $S_2$, and every step in $S_2$ has $\frac{1}{m}$ probability to get reward. If current state is $S_2$, then the expected reward is:

$$\frac{1}{m} \cdot 1 + \left(1 - \frac{1}{m}\right) \gamma \left(\frac{1}{m} \cdot 1 + \left(1 - \frac{1}{m}\right) \gamma \left(\frac{1}{m} \cdot 1 + \left(1 - \frac{1}{m}\right) \gamma (\cdots)\right)\right)$$

$$= \frac{1}{m} \left(1 + \left(\left(1 - \frac{1}{m}\right) \gamma\right) + \left(\left(1 - \frac{1}{m}\right) \gamma\right)^2 + \cdots\right)$$

$$= \frac{1}{1 - \left(\left(1 - \frac{1}{m}\right) \gamma\right)} \frac{1}{m} \overset{def}{=} k$$

So the overall expected reward of the correlated equilibrium is:

$$\mathbb{E}_{cor} = \frac{1}{m} \cdot \gamma k + \left(1 - \frac{1}{m}\right) \gamma \left(\frac{1}{m} \cdot \gamma k + \left(1 - \frac{1}{m}\right) \gamma \left(\frac{1}{m} \cdot \gamma k + \left(1 - \frac{1}{m}\right) \gamma (\cdots)\right)\right)$$

$$= \frac{1}{1 - \left(\left(1 - \frac{1}{m}\right) \gamma\right)} \frac{\gamma k}{m} = \frac{1}{1 - \left(\left(1 - \frac{1}{m}\right) \gamma\right)} \frac{\gamma \frac{1}{1 - \left(\left(1 - \frac{1}{m}\right) \gamma\right)} \frac{1}{m}}{m}$$

In decentralized equilibrium, every step in $S_1$ has $\frac{1}{m^{n-1}}$ probability to transfer to $S_2$, and every step in $S_2$ has $\frac{1}{m^{n-1}}$ probability to get reward. Similarly:

$$\mathbb{E}_{dec} = \frac{1}{1 - \left(\left(1 - \frac{1}{m^{n-1}}\right)\gamma\right)} \frac{\gamma \frac{1}{1-\left(\left(1-\frac{1}{m^{n-1}}\right)\gamma\right)} \frac{1}{m^{n-1}}}{m^{n-1}}$$

Therefore:

$$\frac{\mathbb{E}_{cor}}{\mathbb{E}_{dec}} = \left(\frac{m^{n-1}(1-\gamma)+\gamma}{m(1-\gamma)+\gamma}\right)^2 \geq \left(\frac{m^{n-1}(1-\gamma)}{m(1-\gamma)+m\gamma}\right)^2 = m^{2n-4}(1-\gamma)^2$$

**Detailed derivation of Proposition 2**   Consider the $k$ step game with $k$ states $S_1, \cdots, S_k$. Agents and action spaces are the same as Proposition 1. The discount factor is $\gamma$. $r(s, \mathbf{a}) = 0, s \in \{S_1, \cdots, S_{k-1}\}, \forall \mathbf{a}$; $r(S_k, \mathbf{a}) = \begin{cases} 1 & a_1 = a_2 = \cdots = a_n \\ 0 & \text{otherwise} \end{cases}$. That is, the team can only receive reward in the last step, which is a common case in RL environments. The deterministic state transition function is $T(S_i, \mathbf{a}) = \begin{cases} S_{i+1} & a_1 = a_2 = \cdots = a_n \\ S_i & \text{otherwise} \end{cases}$, $i = 1 \cdots k$, where $S_{k+1}$ denotes the game ending.

Let's prove that $\frac{\mathbb{E}_{cor,k}}{\mathbb{E}_{dec,k}} \geq m^{k(n-2)}(1-\gamma)^k$.

Proposition 1 shows that $\frac{\mathbb{E}_{cor,2}}{\mathbb{E}_{dec,2}} \geq m^{2(n-2)}(1-\gamma)^2$. Now suppose $\frac{\mathbb{E}_{cor,i}}{\mathbb{E}_{dec,i}} \geq m^{i(n-2)}(1-\gamma)^i$, by using similar derivation in Proposition 1, we can get

$$\mathbb{E}_{cor,i+1} = \frac{1}{1 - \left(\left(1 - \frac{1}{m}\right)\gamma\right)} \frac{\gamma \mathbb{E}_{cor,i}}{m}$$

$$\mathbb{E}_{dec,i+1} = \frac{1}{1 - \left(\left(1 - \frac{1}{m^{n-1}}\right)\gamma\right)} \frac{\gamma \mathbb{E}_{dec,i}}{m^{n-1}}$$

So,

$$\frac{\mathbb{E}_{cor,i+1}}{\mathbb{E}_{dec,i+1}} = \frac{m^{n-1}(1-\gamma)+\gamma}{m(1-\gamma)+\gamma} \frac{\mathbb{E}_{cor,i}}{\mathbb{E}_{dec,i}} \geq \frac{m^{n-1}(1-\gamma)}{m(1-\gamma)+m\gamma} \frac{\mathbb{E}_{cor,i}}{\mathbb{E}_{dec,i}}$$

$$= m^{n-2}(1-\gamma)\frac{\mathbb{E}_{cor,i}}{\mathbb{E}_{dec,i}} \geq m^{(i+1)(n-2)}(1-\gamma)^{i+1}$$

**Derivation of Proposition 3**   Given state $s$, since the action space is discrete, suppose each agent has $m$ actions, then we can list all possible joint actions $a_1 \cdots a_n$ in a sequence $\{\mathbf{a}_k\}_{k=1}^{m^n}$. Consider the optimal correlated policy $\pi^*(a_1, \cdots, a_n | s)$. Let $p_k = \pi^*(\mathbf{a}_k | s)$, $k = 1 \cdots m^n$.

Let $Z$ be the value range of $z$. Since $z$ is a continuous random variable, we can divide the support set of $Z$ into $m^n$ disjoint subset $\{Z_k\}_{k=1}^{m^n}$, which satisfies $P(z \in Z_k) = p_k$.

Next, let each agent performs the following deterministic policy:

$$\mu_i(s, z) = (\mathbf{a}_k)_i \text{ if } z \in Z_k$$

That is, agent $i$ performs action $(\mathbf{a}_k)_i$ if $z \in Z_k$. All agents will perform joint action $\mathbf{a}_k$ if $z \in Z_k$, and the team will perform joint action $\mathbf{a}_k$ with probability $P(z \in Z_k) = p_k$. This is exactly the optimal correlated policy.

### A.3   SMAC MAP AND AGENT SELECTION

In SMAC environment, not all maps are suitable for evaluating out method. Because:

- In some map(s), QMIX's performance is not good in non-robust settings (e.g. 3s5z_vs_3s6z, 6h_vs_8z, 27m_vs_30m, corridor). These results can be found in SMAC paper (Samvelyan et al., 2019))

| Map and Agent | 8m
Agent 4 | 8m
Random Agent | 2s3z
Agent 2 | 2s3z
Random Agent | 3m
Agent 2 |
|---|---|---|---|---|---|
| Adv Eval Rate | 0.5 | 0.5 | 0.5 | 0.5 | 0.3 |
| NP | 1.9 | 36.9 | 5.7 | 45.6 | 17.6 |
| NG | 9.9 | 50.9 | 5.6 | 59.6 | 22.9 |
| Gap | 8.0 | 14.0 | -0.1 | **14.0** | 5.3 |
| VA | 53.8 | 59.4 | 78.5 | 60.9 | 32.3 |
| GM | 68.2 | 79.6 | 90.5 | 67.5 | 42.6 |
| Gap | **14.4** | **20.2** | **12.0** | 6.6 | **10.3** |

Table 2: Additional comparison of NG/NP and GM/VA

- In some map(s), the policy trained by QMIX in non-robust settings is already robust with most agents (e.g. so_many_banelines, 3s_vs_{3,4,5}z, 1c3s5z except agent Colossi, MMM).

- In some hard map(s), it may be difficult for agents to achieve robust policy. (e.g. 8m_vs_9m or 10m_vs_11m, our team already has less agents than the enemy; 1c3s5z agent Colossi, Colossi is a large agent, thus if it makes mistakes, other small agents are hard to make up.)

Since big map may more likely be robust when only one agent makes mistakes, we decide to focus on small maps. By examining several maps about QMIX's performance and robustness, we choose four maps to evaluate our method: 8m, 2s3z, 3m, 3s5z.

Since many agents in these maps are homogeneous, we only select some representative agents to evaluate.

- 3m map: All agents are homogeneous. We select the agent that is the least robust one in normal QMIX policy: agent 2. (The test winning rates of normal QMIX policy with 1000 episodes are 18.9%, 21.2%, **17.6%** respectively, when agent 0, 1, 2 has 30% probability to select the worst action.)

- 8m map: All agents are homogeneous. Since this map has more agents than 3m, we select the most robust agent and the least robust agent in normal QMIX policy: agent 4 and agent 6. (The test winning rate of normal QMIX policy with 1000 episodes are 2.3%, 3.4%, 5.4%, 10.3%, **1.9%**, 7.3%, **14.4%**, 7.1% respectively, when agent 0, 1, 2, 3, 4, 5, 6, 7 has 50% probability to select the worst action.)

- 3s5z map: During training we found that this map is difficult to train. We have to train more steps to make it converge, and the training process is slow. Therefore we only evaluate the least robust agent in normal QMIX policy: agent 3. (The test winning rate of normal QMIX policy with 1000 episodes are 27.5%, 31.5%, 80.5%, **9%**, 9.5%, 14.5%, 18%, 22.5% respectively, when agent 0, 1, 2, 3, 4, 5, 6, 7 has 50% probability to select the worst action.)

- 2s3z map: During training we found that this map is a bit difficult to train. We want to only evaluate the least robust agent in normal QMIX policy. But in this map, agent 2 and agent 4 are almost the same robust, so we evaluate both of them. (The test winning rate of normal QMIX policy with 1000 episodes are 6.1%, 8.1%, **5.7%**, 6.4%, **5.7%** respectively, when agent 0, 1, 2, 3, 4 has 50% probability to select the worst action.)

## A.4 ADDITIONAL EXPERIMENT RESULTS

In this section, we present the additional experimental results, including the mean winning rate during training (Figure 4a), the adversarial testing results (Figure 4b), the comparison of NG/NP and GM/VA (Table 2), and the random testing results (Figure 4c).

The results shows that, the performance of GM is better than VA and GV in most settings, except for 2s3z's random agent, in which GV is better than VA but GM is similar to GV. Also, in most settings, the performance improvment of NG/NP is less than GM/VA.

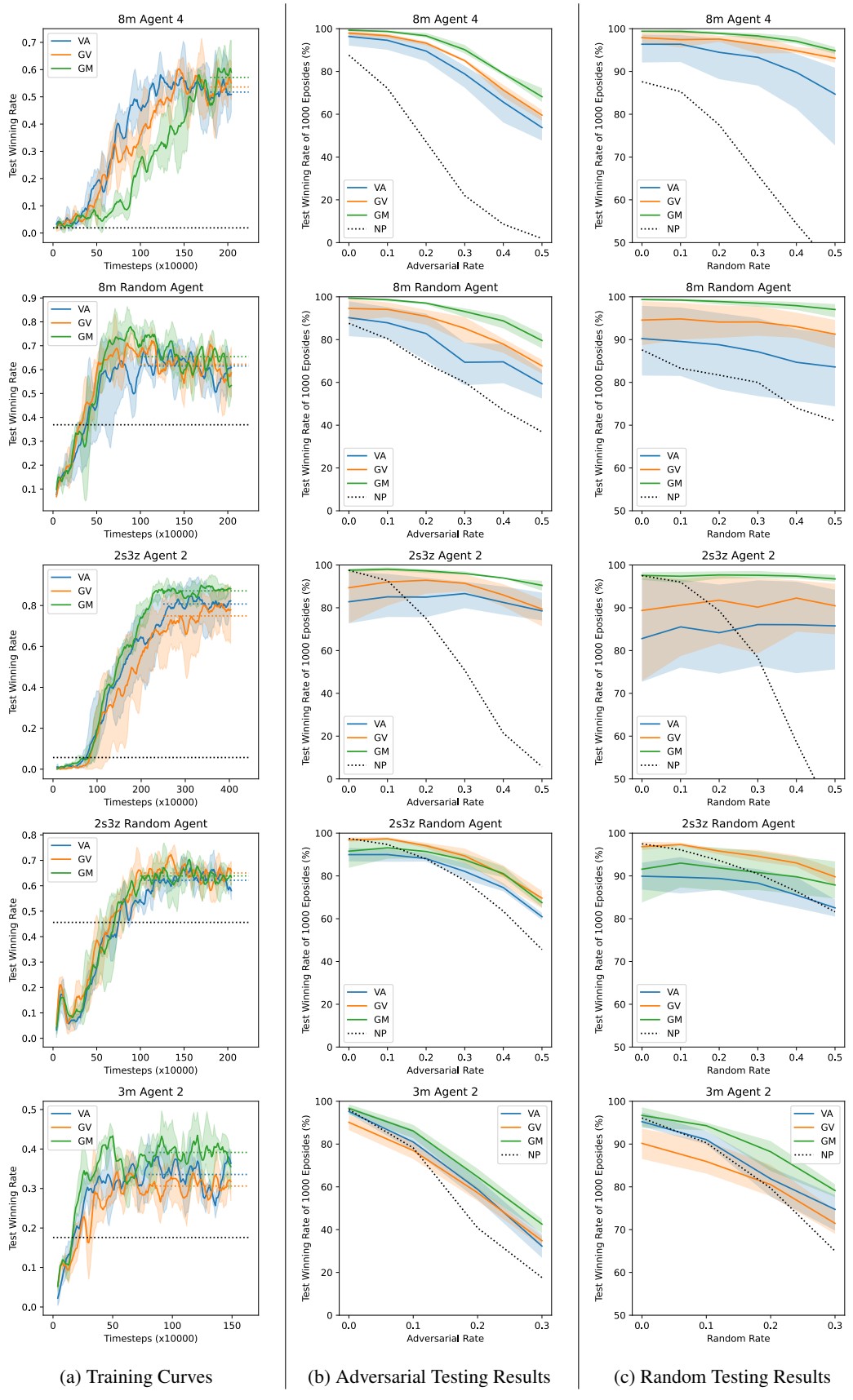

(a) Training Curves     (b) Adversarial Testing Results     (c) Random Testing Results

Figure 4: Additional Testing Results

Table 3: Adversarial Testing Results: Mean test winning rate

| | | | 0% | 10% | 20% | 30% | 40% | 50% |
|---|---|---|---|---|---|---|---|---|
| **8m Map** | Agent 4 | NP | 87.6 | 72.1 | 47.1 | 22.0 | 8.6 | 1.9 |
| | | VA | 96.4 | 94.6 | 89.5 | 78.8 | 65.7 | 53.8 |
| | | GV | 97.9 | 96.7 | 93.3 | 85.0 | 71.3 | 59.6 |
| | | GM | 99.3 | 98.7 | 96.7 | 90.2 | 79.1 | 68.2 |
| | Agent 6 | NP | 87.6 | 79.7 | 67.0 | 53.3 | 31.2 | 14.4 |
| | | VA | 95.4 | 93.1 | 87.2 | 74.8 | 59.4 | 50.2 |
| | | GV | 92.7 | 90.8 | 84.6 | 74.9 | 62.7 | 52.1 |
| | | GM | 98.0 | 96.0 | 91.5 | 84.0 | 73.7 | 63.6 |
| | Random Agent | NP | 87.6 | 80.4 | 68.7 | 60.1 | 47.0 | 36.9 |
| | | VA | 90.2 | 87.9 | 82.8 | 69.4 | 69.6 | 59.4 |
| | | GV | 94.6 | 94.1 | 90.9 | 85.3 | 77.9 | 67.7 |
| | | GM | 99.4 | 98.7 | 97.1 | 93.1 | 88.8 | 79.6 |
| **2s3z Map** | Agent 2 | NP | 97.5 | 92.7 | 75.0 | 50.8 | 21.4 | 5.7 |
| | | VA | 82.8 | 85.1 | 85.0 | 86.6 | 82.5 | 78.5 |
| | | GV | 89.4 | 92.0 | 92.9 | 91.4 | 86.0 | 79.4 |
| | | GM | 97.6 | 98.0 | 97.3 | 96.0 | 93.9 | 90.5 |
| | Agent 4 | NP | 97.5 | 85.7 | 65.0 | 39.1 | 20.9 | 5.7 |
| | | VA | 68.6 | 70.7 | 72.7 | 72.2 | 64.9 | 53.9 |
| | | GV | 92.0 | 91.4 | 88.2 | 81.9 | 71.3 | 58.5 |
| | | GM | 93.4 | 93.0 | 92.5 | 90.2 | 81.6 | 72.6 |
| | Random Agent | NP | 97.5 | 94.7 | 87.9 | 77.9 | 63.5 | 45.6 |
| | | VA | 89.9 | 90.0 | 88.1 | 82.1 | 74.5 | 60.9 |
| | | GV | 96.8 | 97.4 | 94.1 | 89.4 | 80.6 | 69.6 |
| | | GM | 91.6 | 93.1 | 91.4 | 87.5 | 81.1 | 67.5 |
| **3s5z Map** | Agent 3 | NP | 94.4 | 90.7 | 78.7 | 56.5 | 28.6 | 9.0 |
| | | VA | 78.8 | 76.7 | 71.6 | 66.2 | 56.8 | 45.7 |
| | | GV | 79.4 | 67.9 | 68.3 | 66.2 | 60.6 | 51.3 |
| | | GM | 88.9 | 88.1 | 87.2 | 83.4 | 78.9 | 73.5 |
| **3m Map** | Agent 2 | NP | 96.1 | 78.2 | 40.8 | 17.6 | - | - |
| | | VA | 95.2 | 81.0 | 58.8 | 32.3 | - | - |
| | | GV | 90.2 | 77.0 | 57.2 | 34.8 | - | - |
| | | GM | 96.7 | 86.1 | 64.7 | 42.6 | - | - |
| | Random Agent | NP | 96.1 | 80.6 | 56.3 | 32.8 | - | - |
| | | VA | 96.1 | 86.0 | 68.0 | 44.4 | - | - |
| | | GV | 97.1 | 86.1 | 67.2 | 44.2 | - | - |
| | | GM | 97.8 | 90.6 | 72.6 | 51.0 | - | - |

## A.5 SMAC EVALUATION TABLES

The mean testing winning rates of adversarial testing results are shown in Table 3. The mean testing winning rates of random testing results are shown in Table 4.

Table 4: Random Testing Results: Mean test winning rate

|  |  |  | 0% | 10% | 20% | 30% | 40% | 50% |
|---|---|---|---|---|---|---|---|---|
| 8m Map | Agent 4 | NP | 87.6 | 85.3 | 77.5 | 65.8 | 54.4 | 43.9 |
|  |  | VA | 96.4 | 96.4 | 94.5 | 93.3 | 89.8 | 84.7 |
|  |  | GV | 97.9 | 97.4 | 97.6 | 96.3 | 94.8 | 93.1 |
|  |  | GM | 99.4 | 99.4 | 98.9 | 98.3 | 97.1 | 94.8 |
|  | Agent 6 | NP | 87.6 | 87.2 | 85.5 | 83.8 | 83.1 | 75.8 |
|  |  | VA | 95.8 | 94.5 | 93.5 | 91.2 | 87.5 | 83.9 |
|  |  | GV | 92.7 | 93.0 | 91.7 | 90.2 | 87.9 | 81.5 |
|  |  | GM | 98.0 | 96.9 | 96.4 | 94.7 | 93.8 | 90.5 |
|  | Random Agent | NP | 87.6 | 83.3 | 81.7 | 80.0 | 74.0 | 71.0 |
|  |  | VA | 90.2 | 89.6 | 88.8 | 87.2 | 84.7 | 83.6 |
|  |  | GV | 94.6 | 94.8 | 94.1 | 94.1 | 93.0 | 91.3 |
|  |  | GM | 99.4 | 99.2 | 98.9 | 98.5 | 97.9 | 97.0 |
| 2s3z Map | Agent 2 | NP | 97.5 | 96.0 | 89.3 | 78.4 | 58.6 | 40.5 |
|  |  | VA | 82.8 | 85.5 | 84.2 | 86.1 | 86.0 | 85.8 |
|  |  | GV | 89.4 | 90.6 | 91.8 | 90.1 | 92.3 | 90.5 |
|  |  | GM | 97.6 | 97.4 | 97.7 | 97.6 | 97.4 | 96.7 |
|  | Agent 4 | NP | 97.5 | 94.4 | 86.1 | 78.4 | 66.3 | 52.6 |
|  |  | VA | 68.6 | 68.3 | 68.6 | 68.6 | 68.9 | 69.7 |
|  |  | GV | 92.0 | 85.1 | 86.1 | 81.8 | 79.5 | 76.7 |
|  |  | GM | 93.4 | 92.8 | 92.9 | 92.6 | 91.7 | 91.1 |
|  | Random Agent | NP | 97.5 | 96.1 | 93.6 | 90.5 | 86.4 | 81.6 |
|  |  | VA | 89.9 | 89.7 | 89.4 | 88.3 | 85.5 | 82.5 |
|  |  | GV | 96.8 | 97.3 | 95.8 | 94.6 | 93.0 | 89.8 |
|  |  | GM | 91.6 | 93.0 | 91.9 | 90.8 | 89.8 | 87.9 |
| 3s5z Map | Agent 3 | NP | 94.4 | 92.9 | 86.0 | 79.6 | 68.5 | 57.7 |
|  |  | VA | 78.8 | 76.2 | 76.3 | 73.6 | 69.4 | 65.7 |
|  |  | GV | 79.4 | 66.1 | 64.9 | 65.6 | 63.3 | 62.4 |
|  |  | GM | 88.9 | 89.7 | 88.0 | 87.6 | 86.3 | 83.8 |
| 3m Map | Agent 2 | NP | 96.1 | 90.3 | 79.7 | 65.0 | - | - |
|  |  | VA | 95.2 | 91.0 | 81.9 | 74.7 | - | - |
|  |  | GV | 90.2 | 85.9 | 80.5 | 71.5 | - | - |
|  |  | GM | 96.7 | 94.3 | 88.2 | 79.1 | - | - |
|  | Random Agent | NP | 96.1 | 92.2 | 84.6 | 74.1 | - | - |
|  |  | VA | 96.1 | 92.3 | 87.1 | 80.4 | - | - |
|  |  | GV | 97.1 | 94.1 | 87.6 | 81.6 | - | - |
|  |  | GM | 97.8 | 96.1 | 91.0 | 87.7 | - | - |

