# OpenReview forum: "Robust Multi-Agent Reinforcement Learning Driven by Correlated Equilibrium"
_ICLR.cc/2021/Conference — Reject_

### Official Review · AnonReviewer1 · 2020-10-26
**Blind review**

**Rating:** 5
**Confidence:** 4

**Review:**

This paper describes a new method for making cooperative MARL algorithms more robust to teammate mistakes. The approach learns correlated equilibria that depend on a global random variable. The setting is discussed with some theoretical results and experiments are provided to compare versions of their approach in Starcraft (SMAC).

Safe and robust MARL is an important topic. There may be settings in which agents make mistakes or can be in dangerous situations. Making agent policies robust to such circumstances will be necessary in real-world scenarios. Also, the idea of using mutual information to improve agent performance is promising. Sometimes, such information is available (or can be available cheaply) so why not take advantage of it?

There are some questions about the exact robust MARL setting that is approached in this paper. In particular, the paper assumes the setting where only one agent makes a mistake at each timestep and that mistake-making agent is adversarial (i.e., chooses the worst possible action for the team). The specific setting in Eq 5 has the agent choose the minimizing action with a small probability and the maximizing action with the remaining probability. When is such a setting realistic? If all the agents are assumed to be trained centrally with all the agent policies known, why would an agent become adversarial in this way? The setting should be motivated to better understand how realistic and general it is.

The theoretical results are extensions of the normal form case to the stochastic game case. The theory appears novel and as a result useful, but not a major contribution.

The algorithm is somewhat straightforward, but there are novel aspects and the idea is promising. As the paper points out, other approaches have used global random variables, but not in this robust context and not with the form of mutual information that is added to improve the use of the global signal.

The experimental results show the method works well, but they are not sufficiently comprehensive. For example, how does it perform if *no* agents make mistakes? This appears to be shown in Figure 2, but the results aren't clear. It seems like sometimes the proposed methods do better than the normal policy. This shouldn't be possible in the fully observable case, but may be due to the fact that SMAC is partially observable. Using policies that are dependent on a global random variable can improve performance in partially observable settings. This is known (e.g., see the paper below), but never discussed in the paper. SMAC domains are partially observable so it isn't clear how much that is a factor. Because the paper and theory are described from a fully observable perspective, the paper should also have results for the fully observable case. The SMAC results are useful, but the proposed method should also be discussed in terms of partial observability.

Bernstein, Daniel S., et al. "Policy iteration for decentralized control of Markov decision processes." Journal of Artificial Intelligence Research 34 (2009): 89-132.

The writing is generally understandable, but it should be polished. There are some issues that are not clear such as the fact that a policy in the partially observable case (defined in 3.1) could do better by conditioning on the history.

---

> ### Author Response · Authors · 2020-11-23
> **Replies to AnonReviewer1**
>
> Thank you very much for your review of our paper.
>
> Here are our responses to each point you mentioned:
>
> "Motivation of problem formulation"
> - The motivation of our formulation is to obtain a policy that is robust when one agent makes some but not very big mistakes. In normal MARL algorithm, the team can guarantee to achieve high reward only when all agents accurately execute their optimal strategy. However, this may not always be true in real world scenarios. Real world agents may occasionally make mistakes (e.g. machine malfunctioning).
> - Also, we assume that real world agents may **only occasionally** make mistakes, since most people won't deploy a agent that always/frequently makes mistakes in real scenario. That is, agent has some probability ($\varepsilon$ ) of making mistakes. The real problem we want to deal with is to obtain a robust policy when one agent has $1-\varepsilon$ probability to perform its optimal action, and $\varepsilon$ probability to perform randomly mistaken action. To solve this problem we uses adversarial training to optimize its worst case: in the $\varepsilon$ probability agent performs its worst action.
> Thanks for pointing out this and we have already added these clarifications into the revised version.
>
> "More experiment baselines and explanation"
> - In this paper we focused on QMIX algorithm, and we wonder whether QMIX algorithm is suitable for fully observable case, since QMIX algorithm is mainly designed for partially observable environment like SMAC.
> - We tried another way to explain the importance of correlation in robust CMARL settings: we run normal training with global variable and mutual information loss, and then evaluate it under adversarial settings. We found that the improvement of the performance using normal training in most maps is not as high as  that in adversarial training. These results show that "correlation is more important in robust settings". We have already added these results into the revised version. Now the overall thought of our paper is: In fully observable cases, correlation is not important in normal CMARL settings, but it's important in robust CMARL settings; In partially observable case, correlation is somehow useful in normal CMARL settings, but it's more important in robust CMARL settings.

---

### Official Review · AnonReviewer3 · 2020-10-27
**Comments**

**Rating:** 4
**Confidence:** 3

**Review:**

This paper considers a case in the cooperative multiagent reinforcement learning where one single agent can behave adversarially. It claims that the performance of the whole MARL system  would deteriorate significantly  and demonstrate this phenomenon both theoretically and empirically. Then the author proposes a solution, i.e. , solving the correlated equilibrium in the game.  To seek  the correlated equilibrium, the author introduces a global random variable z and adds a mutual information regularization term.  Indeed this idea is easy to follow since it directly follows the definition of the correlated equilibrium. At last, it tests the algorithm in the SMAC environment and compares it with QMIX algorithm.




Question 1: How do you get the global random variable z in Q(o_i, z,a_i) in the execution. Is it sampled from the posterior distribution q(z|o,a)?

Question 2: my main concern is about the novelty of this paper. Can you discuss the difference between your method and MAVEN[1], where the author also introduces the latent variable z and uses the mutual information on z and observed trajectory.  The idea seems to be almost the same especially when it is applied on QMIX.

Question 3: The author should include more baselines. Here I just see the result from QMIX. The author should test this idea on other classical methods, e.g., COMA and many others.

[1] MAVEN: Multi-Agent Variational Exploration

---

> ### Author Response · Authors · 2020-11-23
> **Replies to AnonReviewer3**
>
> Thank you very much for your review of our paper.
>
> Here are our responses to each point you mentioned:
>
> -"Question 1: How do you get the global random variable z in Q(o_i, z,a_i) in the execution. Is it sampled from the posterior distribution q(z|o,a)?"
> - The global variable $z$ is sampled from a uniform distribution independently at each timestep. $z$ itself does not contain any state or environment information, agents just use it as a common knowledge and learn to use it to correlate their policies.
>
> -"Question 2: Difference from MAVEN"
> - The setting of our paper is completely different from MAVEN. In MAVEN, the latent variable $z$ is sampled once per **episode** (see their figure 2 top-left corner), and optimize the mutual information between $z$ and the whole episode's data will lead to a more dispersed episode distribution, which encourages exploration. In our setting, $z$ is sampled every timestep, and optimizing the mutual information between $z$ and each agents' action will urge agent to perform a policy that is related to $z$, therefore urges agents to perform correlated policy.
>
>
> -"Question 3: The author should include more baselines"
> - There 're few works on robust MARL topic. To the best of our knowledge, we are the second one to study the action-robust MARL problem (the first one is M3DDPG [1] which uses vanilla adversarial training). And we are the first one to demonstrate the importance of correlation in this kind of robust MARL setting, from both theoretical and empirical aspects. This is our main novelty. As a first step work, the method itself is straightforward but much effective.
> - For the COMA algorithm, the learning environment of origin COMA paper is not open sourced. Actually we have tried COMA algorithm in SMAC environment, but as in SMAC paper and our experiments, the performance of COMA in SMAC is much worse than QMIX in most maps. Therefore we doubt whether it's worthy of considering COMA in robust MARL situation in SMAC environment, if the normal performance is not good. This is also the reason that we only evaluate the robustness in the maps that QMIX can have high performance in normal settings.
>
> [1] Li, Shihui, et al. "Robust multi-agent reinforcement learning via mini-max deep deterministic policy gradient." Proceedings of the AAAI Conference on Artificial Intelligence. Vol. 33. 2019.

---

### Official Review · AnonReviewer2 · 2020-10-28
**Interesting problem, but vague writing and insufficient experimental proof**

**Rating:** 3
**Confidence:** 4

**Review:**

Summary

Robustness in the multi-agent setting is a nuanced concept, as (a subset) of agents can act adversarially, while the non-adversarial agents can be trained to be more robust. The authors propose to solve a max-min problem, in which some agents are optimizing their rewards taking into account some agents maybe acting suboptimally/adversarially.

The authors implement learning using QMIX and evaluate on SMAC. They train a model p(z | a, o) for a global latent variable (using a variational lower bound on mutual info between z and a) that each robust agent conditions its policy on.

The authors claim this method can yield significant performance improvements over vanilla adversarial training.


Strengths

- Robustness is not well explored in the multi-agent setting, and using a correlated equilibrium seems like an interesting way to model coordinated robust training for (a subset) of MARL agents.

Weaknesses

Throughout, the writing is not very clear.

- The notation in (3) and (4) is sloppy: which indices are "team" and which ones are "mis"?
- Sect 4.1 talks about differences between correlated eq and decentralized equilibria, but the authors never formally define what the difference is between "cooperative agents" (which they claim emerges under centralized training, decentralzied execution), and a correlated equilibrium (in which some agents cooperate, presumably, but others don't). I assume the authors are getting at the normal definition of correlated eq: there is some (global) random variable that the policies are conditioned on, but the text is very unclear about this.
- The propositions 1 - 2 seem out of place without a formal defn of correlated eq (see above).
- The text around prop 3 never mentions that agents presumably now learn a policy pi(a | s, z). This should be clarified.
- Eq 7 is missing indices i, or should clarify the "a" refers to a joint action, etc.

Conceptual issue:

- Adversarial agents are defined as taking actions that minimize their *own* Q-value (i.e., agent i executes argmin_i Q_i). But shouldn't the worst mistake be the action that minimizes other agents' value? It seems this definition of "adversarial" agent is rather weak, and counter to what is defined in Eq (1 - 3), where adversarial agents want to minimize other agents' value as well? This is where the notation again causes confusion.

The experimental results are too thin to conclude the authors' method is indeed effective.

- What is the significance (if any) of having different agents (6, 4, etc) as the "adversarial" ones in Figures 1, 3?
- The authors do not explain what the structure of SMAC is (the experiment environment).
- The authors should visualize the distribution of actions executed by the robust/adversarial agents to give some intuition of the qualitative differences.

Questions

- What is a "flat maximum" (above Eq 3)?

---

> ### Author Response · Authors · 2020-11-23
> **Replies to AnonReviewer2**
>
> Thank you very much for your review of our paper.
>
> Here are our responses to each point you mentioned:
>
> "Writing is not very clear."
> - We have already modified our revised paper and correct these writing.
> - eq (3) and (4) is removed.
> - In prop 3 agents are learning deterministic policy $mu_i(s,z)$, it is actually $pi_i(a_i|s,z)$ but writing in different notation.
> - Since the sentence above eq 7 mentioned $I(z_t;a_{it}|o_{it}), so we omit the indices i. We will clarify this omission more clearly.
>
> "Formally definition of 'correlated' and 'decentralized'"
> - In this setting, we just define the correlated equilibrium as the optimal joint policy, and the decentralized equilibrium as the optimal disjoint policy. We have already added these definition into the revised paper, and correct some notations and writings.
>
> "Adversarial agent minimizes its own Q value, and this is a rather weak adversarial since it doesn't use other agent's information"
> - The motivation of our formulation is to obtain a policy that is robust when one agent makes some but not very big mistakes. In normal MARL algorithm, the team can guarantee to achieve high reward only when all agents accurately execute their optimal strategies. However, this may not always be true in real world scenario. Real world agents may occasionally make mistakes (e.g. machine malfunctioning).
> - But these mistakes are most likely **not related** to the team goal and other agents' policy. In real world scenarios, agents may occasionally make mistakes **randomly**. We think that, the case that one agent uses the team goal and other agents' policy to **deliberately** making trouble to the team is rare in real world scenario. So we mainly consider the case that the adv agent's mistake is not related to the team goal and other agents' policy, and following this consideration, we just simply let the adversarial agent execute the action that minimizes its own Q-value. This is a weak adversarial, but it do cover most real world robustness cases. The real adversarial case is challenge, since it contains two MDPs that nested with each other. So as a first step work we only solve a weaker and simpler case, and we will consider this challenge in our future work.
>
> "The significance of having different agents as the 'adversarial' ones"
> - We actually explain this in appendix. For the "fixed agent" case, since many agents in these maps are homogeneous, we only select some representative agents for evaluation. We mainly consider agents that are the least robust ones based on the policy trained in normal environment.
>
> "Explain what the structure of SMAC is"
> - The SMAC environment we uses is exactly the same as the one in QMIX paper, and we cite QMIX. We added some brief description of SMAC environment in our experiment section.
>
> "What is a 'flat maximum'"
> - Flat maximum is that, the expected reward will not dramatically decrease when the policy changed only a little bit. In our revised paper we no longer uses this concept.

---

### Official Review · AnonReviewer4 · 2020-10-28
**Clear exposition, sound experiments, some details missing**

**Rating:** 6
**Confidence:** 3

**Review:**

## Overview

This paper considers robustness in the context of multi-agent setting with decentralized execution, in the scenario where one agent may behave sub-optimally (either by being adversarial or simply taking random actions) on a fraction of the time-steps.

The solution relies on attempting to find a correlated equilibrium by providing agents with a common source of randomness.

The paper then exposes some experimental results on Starcraft. The experiments seem sound and prove the point. However, some details seem to be missing, potentially raising concerns for reproducibility, and the exposition could potentially be improved for better clarity.
Despite these short-comings, which should be easily addressable, the results are convincing and the simplicity of the approach warrants its acceptance.

## Method

The paper starts by theoretically justifying the need of finding a correlated equilibrium, and thus do a good job at motivating the approach.

The method relies on augmenting the agents by providing access to a shared source of entropy, in the form of a shared low dimensional random variable $z$.

To encourage the network to rely on this $z$, the authors add a variational-lower bound type of loss that experimentally helps the performance.

## Experiments

The authors carry out experiments on some Starcraft environments of the SMAC suite.
They build on an established algorithm for these environments, namely QMIX, and provide a reasonable "vanilla" adversarial training scheme as baseline.

Globally, the full method using the variational lower bound loss seems to be performing consistently better than both the adversarial training baseline and the version of the training with only access to z but no variational lower bound loss.

The reporting of the results could probably be improved:
* Figure 2 and 3 use different vertical axis range, making direct comparison difficult
* Training curves are indeed interesting to give an intuition of the general behavior, but I think it would be valuable to provide in the main text an aggregated table summarizing the results in table 1 and table 2 from the appendix, with proper confidence intervals (something like an average performance over maps and agents). This would better convince the reader that the better performance of QMIX-GM is a general trend in all the possible settings.


The paper specifies that the method uses a 3-dimensional uniform distribution in $[0,1]^3$ for $z$, however the design decisions behind such a choice are not discussed or experimentally backed. In particular, it is not clear if the dimensionality of this variable plays any role in the final performance, and if/how practitioners may have to tune this knob when applying the method to their own problem.

Finally, some details are lacking in the architecture used:
* The text says that the latent code $z$ is added as input to the Q network, but doesn't specify the architectural changes that are made to made this possible.
* As for the variational approximation, the text reads "[we] apply a neural network to output its mean and variance". However, not details are provided on this network, and in particular whether it shares parameters with the aforementioned Q network

---

> ### Author Response · Authors · 2020-11-23
> **Replies to AnonReviewer4**
>
> Thank you very much for your review of our paper.
>
> Here are our responses to each point you mentioned:
>
> "Improvements of reporting the experimental results"
> - Since figure 2 and  3 are for testing different scenarios, and their performance is largely different. The adversarial evaluation's performance is significantly lower than random evaluation. We found that if we choose 0-100 as the vertical axis of figure 3, then the curve may become too flat and make it harder to notice the improvement of global variable with mutual information method.
> - Table 1 and Table 2 in the appendix is actually the numerical result of Figure 2 and Figure 3.
>
> "Design decisions behind choose 3-dim $z$"
> - In this work we just simply consider the dimensional of $z$ as a hyperparameter, and we select it based on its empirical  performance.
>
> "Details lacking in the architecture"
> - To add latent variable $z$ into the model, we just extend the agent's observation space from `smac_dim` to `smac_dim+3`. At each timestep, we sample a 3-dim random variable and put it into the last 3 dimension of all agents' observation.
> - The neural network we uses to learns the mean and variance of the variational posterior is another network, and does not share parameters between the Q network. We use a 2-layer dense network with 32 unit and ReLU activation. The network takes agent's observation (without latent variable) and agent's action as input, and output a 6-dim vector with first 3 dim as mean and last 3 dim as variance.
> - We have already added these explanations in the revised paper.

---

### Official Review · AnonReviewer5 · 2020-11-09

**Rating:** 4
**Confidence:** 3

**Review:**

Summary: The paper addresses the issue of robustness in cooperative multi-agent RL setups, where the inclusion at test time of an agent that makes error or is even adversarial can drastically decrease performance. The main idea is to compute a correlated equilibrium, by allowing all agents policies to depend on a common signal. To encourage the actions of the agents to correlate, they add a mutual information loss (i.e. a retrodiction that encourages the global latent to be predictable given the action taken).

High level review:
The paper introduces a natural solution to the studied problem. However, it suffers from a number of flaws that precludes acceptance at ICLR in my mind.

- The proposed method is extremely similar to previous work (Chen et. al and Kim et. al). In particular Chen et al. also suggest something essentially equivalent to the Infomax loss (see equation 5). Their particular definition of the distribution is solely needed for the theoretical proof that correlated policies can be obtained by having a global latent variable (theorem 2.2 in their paper, which is essentially equivalent to proposition 3 in this paper; both theorems are fairly standard use of latent variables to represent correlated distributions). I fail to appreciate the genuine contribution of this paper over Chen et. al. Note that neither approaches (Chen et. al, Kim et. al) are used as baselines in this paper, while they clearly should be.

- The writing is poor and can often be very confusing.

- The theoretical section does not connect well to the rest of the paper ; section 3 is almost entirely disconnected from what the authors suggest doing (equation 4 is effectively not solved at all, so why introduce this hierarchy of relaxation if the proposed method only poorly relates to it? In the end, the authors train instead an agent in a regular MARL setup with a global variable, and test it in the robustness setting; this is a heuristic that entirely replaces the maximin approach of equation 4 as far as I can understand). Is the counterexample in section 4 novel, or folklore? (It feels very familiar). I would not couch it in terms of a proposition (which is oddly informal - 'can be much larger' - for a theoretical statement), and instead study it more carefully as an intuitive example for why correlation is needed.  A team could play in a decentralized way and obtain very good reward by using the (1,1,1,1,1) action with probability one (which does not require correlation). The issue is that this strategy is not robust to adversaries. This should be explained in more detail.

Reviewer familiarity with the work: I am moderately familiar with the MARL literature (MADDPG, COMA; I worked on on MARL paper), and with game theory (correlated equilibrium). The precise setting of the paper I was not familiar with (robustness with respect to switching from cooperative to mix-cooperative; QMIX algorithm). I was not aware of the work of Kim et al.

---

> ### Author Response · Authors · 2020-11-23
> **Replies to AnonReviewer5**
>
> Thank you very much for your review of our paper.
>
> Here are our responses to each point you mentioned:
>
> "Similarity with to previous works (Chen et. al and Kim et. al)"
> - The method itself is similar to previous works you mentioned (and we cite them). The main difference is the problem settings: their work is just to use global variable to encourage correlation in normal MARL situation, and we are using global variable to encourage correlation to improve the performance in robust MARL setting.
> To the best of our knowledge, we are the first to demonstrate the importance of correlation in this kind of robust MARL setting, from both theoretical and experimental aspects. This is our main contribution. So as a first-step work, we just use the most straightforward method to demonstrate this important finding. Therefore the main purpose of our work is actually providing a first valid solution to the robust cooperative MARL. Indeed, using some more complex correlation method like these two works or other in robust MARL situation is a research directions that is worth considering, and we prefer to do this in our future works, and this work's result could be acted as a baseline.
>
> "Section 3 is almost entirely disconnected from latter section"
> - The connection between section 3 and latter section is weak, we agree with you. The problem shows in section 3 is the ultimate goal of this kind of robust MARL. But it's hard to solve, so as a first step work, we solve a simpler version. We modify our section 3 and put most content their into last `discussion and future work` section.
>
> "Explanation of the counterexample in section 4"
> - The counterexample is actually inspired by other team minimax work on normal form games or extensive form games, and we extend it into stochastic games / MDPs.
>
> "Why decentralized policy {1,1,1,1,1} is not robust"
> - Consider a situation that one agent will make some mistakes itself (i.e. it does not correlate its mistaken policy with the team's policy). If the team performs {1,1,1,1,1} with probability one without correlation, then if the mistaken policy takes action 1 with less than 1/m probability, then the team's expected reward will be less than 1/m. However, if the team performs {1,1,1,1,1}...{m,m,m,m,m} with probability 1/m with correlation, then whatever the mistaken policy is (as long as it's independent
> of the team's policy), the team will still get 1/m expected reward.

---

### Decision · Program_Chairs · 2021-01-07
**Final Decision**

**Decision:**

Reject

**Comment:**

The paper tackles the interesting area of cooperative multi-agent learning and presents a promising method to make MAL robust to mistakes of teammates, while learning correlated equilibria. Reviewers find the presented setting and theoretical contributions limited and the experiments not extensive enough; also some technical details about the architecture are lacking, and the notation and writing can be substantially improved upon. As such the paper does not seem ready for publication at this stage.